# Genetic Diversity of *Actinobacillus pleuropneumoniae* Serovars in Hungary

**DOI:** 10.3390/vetsci9100511

**Published:** 2022-09-20

**Authors:** Gábor Kardos, Rita Sárközi, Levente Laczkó, Szilvia Marton, László Makrai, Krisztián Bányai, László Fodor

**Affiliations:** 1Institute of Metagenomics, University of Debrecen, Nagyerdei krt. 94, H-4032 Debrecen, Hungary; 2Department of Microbiology and Infectious Diseases, University of Veterinary Medicine, Hungária krt. 23–25, H-1143 Budapest, Hungary; 3Veterinary Medical Research Institute, Hungária krt. 21, H-1143 Budapest, Hungary

**Keywords:** *Actinobacillus pleuropneumoniae*, toxin gene profile, pulsed-field gel electrophoresis, whole-genome sequencing, clonality, antibiotic resistance

## Abstract

**Simple Summary:**

*Actinobacillus pleuropneumoniae* causes severe pneumonia in pigs, resulting in high economic losses. A total of 114 isolates from pneumonia were characterized by the examination of biotype, serovar, antibiotic resistance genes, and genes of toxin production. Analyzing their genetic relationship, 16 groups of related isolates were found. The genetic diversity was different in the different groups, however. It was remarkably small in the case of serovar 13, which was unusually frequent in Hungary. Therefore, representative isolates of serovar 13 were subjected to whole-genome sequencing, confirming low diversity. Antibiotic resistance was frequently found in isolates of serovar 13 but was less frequent in other serovars. The unusually high frequency and low diversity of serovar 13 suggest a clonal spread in Hungary, which may have been facilitated by a high frequency of resistance to beta-lactams and tetracyclines.

**Abstract:**

A total of 114 *Actinobacillus pleuropneumoniae* isolates from porcine hemorrhagic necrotic pleuropneumonia were characterized by the examination of biotype, serovar, antibiotic resistance genes, and genes of toxin production. Pulsed-field gel electrophoresis was used to analyze their genetic relationship, which identified 16 clusters. Serovar 2 (50 isolates), serovar 13 (25 isolates), serovar 9 (11 isolates), and serovar 16 (7 isolates) were the most frequent serovars. Serovar 2 formed nine distinguishable clusters; serovar 13 and serovar 16 were less diverse, exhibiting two potentially related subclusters; serovar 9 was represented by a single cluster. Remarkably small differences were seen in the core genome when nine representative isolates of serovar 13 were subjected to whole-genome sequencing. Tetracycline resistance was relatively frequent in the two clusters of serovar 13; one of them was also frequently resistant against beta-lactams. Resistance in other serovars was sporadic. All isolates carried the *apxIV* gene. The toxin profiles of serovar 2 were characterized by the production of ApxII and ApxIII toxins, except for a small cluster of three isolates: serovar 9 and serovar 16 isolates produced ApxI and ApxII toxins. Serovar 13 carried *apxII* and *apxIBD* genes, indicating the production of the ApxII toxin, but not of ApxI or ApxIII. The unusually high frequency and low diversity of serovar 13 are not explained by its virulence properties, but the high frequency of resistance to beta-lactams and tetracyclines may have played a role in its spread. The emergence of serovar 16 may be facilitated by its high virulence, also explaining its high clonality.

## 1. Introduction

*Actinobacillus pleuropneumoniae* is a major pathogen of swine. Two biotypes and 19 serovars (serotypes) can be differentiated within the species; however, sometimes nontypable strains can also occur. Strains of biotype I need nicotinamide adenine dinucleotide (NAD, V-factor) to culture, while biotype II strains can replicate without it [1,2,3,4,5]. *A. pleuropneumoniae* can infect domestic pigs but occasionally it can also colonize wild boars [6].

Different virulence variants of the agent have been described. Some strains cause high mortality, while others are intermediate in virulence or avirulent. In addition to type 4 fimbria, outer membrane proteins, capsules, surface polysaccharides, lipopolysaccharide, biofilm formation capacity, extracellular enzymes, and toxins were identified as virulence factors [7,8,9,10,11,12,13,14,15]. The major virulence factors of *A. pleuropneumoniae* are exotoxins, which are RTX toxins (repeats in toxins). *A. pleuropneumoniae* strains can produce four types of exotoxins. The secretion of toxin effectors *apxIA* and *apxIIA* is mediated by the *apxIBD* exporter; the production of ApxIII and ApxIV toxins is independent of *apxIBD* [8,9]. The toxin ApxIV is characteristic of and produced by all isolates of the species [16,17]; four variants of its gene have been found [18], but it remains unknown whether the encoded toxins differ in their function. The production of the other three toxins ApxI, ApxII, and ApxIII varies among serotypes [19,20]. The formation of ApxII toxin is almost universal among serovars, except serovars 10 and 14, though serovar 3 seems to be unable to secrete it due to missing *apxIBD* [18]. Though this toxin is a less potent leucocidin than ApxI or ApxIII, it is capable of synergistically enhancing their effect [21]. Many *A. pleuropneumoniae* strains produce ApxI or ApxIII together with ApxII. ApxI toxin, which is considered to confer the highest virulence [9,22,23], is usually produced by serovars 1, 5a, 5b, 9, 10, 11, 14, and 16 [19,24], and it is a major, though not the sole contributor to lung damage and disease caused by these serovars [9,12,25]. Therefore, serovars producing ApxI are considered more virulent than those serovars that do not produce it [22,23]. Serovars carrying *apxIII* together with *apxII* are serovars 2, 4, 6, 8, and 15; they are also considered virulent, i.e., traditionally, high virulence is attributed to the presence of at least two secreted exotoxins (ApxII plus ApxI or ApxIII) besides the intrinsically produced ApxIV, while biotypes/serovars carrying only one toxin gene besides ApxIV are considered to show low virulence, thus they are unlikely to cause epizootics [3]. In addition to typical toxin-producing strains, atypical ones lacking certain genes also occur [26]. This virulence-based grouping is in line with the lipopolysaccharide O-chain or *omlA* phylogroups as well [3,27]; however, other virulence factors also contribute to disease severity [12]. Most conventional herds are infected with one or more serovars of *A. pleuropneumoniae*, but these strains usually show low virulence and their serovar pattern is different from that of isolates causing severe diseases [3].

*Actinobacillus pleuropneumoniae* shows considerable geographic variation, and differences in the dominant serovars in different countries and regions have been reported [3]. Outbreaks and sporadic cases are frequently caused by different serovars. Serovar 2 was dominant in Germany and Hungary [28,29], it was the second most common in the Czech Republic [30,31], while serovar 8 was the most frequently isolated serovar in England and Wales [32]. Serovars 5 and 7 were the most frequent serovars in Canada [33,34]. Nevertheless, biotype 2 strains are more commonly isolated in Europe, e.g., in Hungary [29,35], Denmark [36], or Spain [37], than in America.

The aim of the study was the investigation of the genetic diversity of *A. pleuropneumoniae* in Hungary using pulsed-field gel electrophoresis and focusing on the correlation between genotypes, serovars, toxin profiles, and the presence of antimicrobial resistance genes.

## 2. Materials and Methods

### 2.1. Samples

A total of 114 *A. pleuropneumoniae* strains isolated between 1995 and 2014 from the lungs of pigs with hemorrhagic necrotic pneumonia were included in the examinations. Samples were collected from pigs that died because of pleuropneumonia or from lung lesions in slaughterhouses. The samples originated from 44 different swine herds located in different parts of Hungary.

### 2.2. Bacterium Culture, Serotyping, Antibiotic Resistance, Toxin Profile

*Actinobacillus pleuropneumoniae* strains were isolated on Tryptone Soya Agar (TSA, Biolab Ltd., Budapest, Hungary) containing 10% defibrinated sheep blood and cross-inoculated with a nurse streak of *Staphylococcus aureus.* They were later cultured on chocolate agar completed with 50 μg/mL NAD (Biolab Ltd., Budapest, Hungary). The agar plates were incubated at 37 °C for 24 h in an aerobic environment with the addition of 5% carbon dioxide.

The *A. pleuropneumoniae* strains were identified and biotyped using standard methods [38,39] and their identity was confirmed by examination with 16S rRNA PCR, as described earlier [6]. The isolated *A. pleuropneumoniae* strains were stored at −80 °C until further examinations.

Serovar typing was performed using an indirect hemagglutination test, as described previously [24,29]. Unfortunately, some archive isolates lost viability before serotyping (Figure 1).

Susceptibility to amoxicillin, penicillin, oxytetracycline, doxycycline, florfenicol, cefoperazone, tiamulin, and enrofloxacin was determined by broth microdilution tests following the guidelines of the Clinical and Laboratory Standards Institute [40]. Resistance genes *bla*_ROB-1_, *bla*_BRO-1_/*bla*_BRO-2_, *tet*A, *tet*B, *tet*L, *tet*H, and *tet*K were detected by PCR assays [41,42,43,44]. Toxin profiling was performed as described [20].

### 2.3. Pulsed-Field Gel Electrophoresis

Pulsed-field gel electrophoresis was used to assess genetic relatedness. Plug preparation was performed as described earlier [45], plugs were digested with *Apa*I, and macrorestriction profiles were obtained in a CHEF DRIII instrument with electrophoresis parameters following Chevallier et al. [46]. Profiles were analyzed using the software Fingerprinting II (BioRad, Hercules, CA, USA) using the Dice similarity coefficient and the UPGMA clustering method. The threshold for probable relatedness was set at 85%.

### 2.4. Whole-Genome Sequencing

Genomes of nine serotype 13 isolates were chosen for whole-genome sequencing to represent the two pulsotypes and various isolation years and herds of origin. Sequencing was performed using the Illumina NextSeq 500 platform to obtain 150 bp-long single-end reads. Read quality was checked by FastQC 0.11.9 [47] to ensure sufficient data quality, then reads were assembled using SPAdes 3.14.1 [48] with default options. The contiguity of assemblies was checked using QUAST 5.0.2 [49] and completeness was assessed using BUSCO 5.2.2 [50]. The pangenome of samples was reconstructed using Panaroo 1.2.10 [51], a pipeline that ensures the accurate identification of gene presence/absence by clustering and re-finding genes in genome sequences annotated by Prokka 1.14.6 [52]. The aligned core genes were subject to phylogenetic reconstruction using IQ-TREE 1.6.12 [53] using the automatic model selection (-m MFP) and testing the statistical robustness (aLRT and ultrafast bootstrap) using 1000 replications. The phylogenetic tree and the gene presence/absence matrix were plotted using the R 4.0.4 [54] packages ggtree [55] and ggplot2 [56]. The blocks of gene clusters unique to given lineages were filtered using the awk 4.1.4 programming language and the functional re-annotation of these genes was carried out using the eggNOG mapper [57] webserver (http://eggnog-mapper.embl.de/, accessed on 21 July 2022). The identity of phage sequences was checked using the PHASTER webserver (https://phaster.ca/, accessed on 21 July 2022). Resistance genes were screened using ABRicate (Seemann, T; https://github.com/tseemann/abricate, accessed on 21 July 2022).

## 3. Results

### 3.1. Serovars

The predominance of serovar 2 was evident in 50 isolates from at least 26 different geographically and technologically isolated herds. Further major serovars were serovar 13 (25 isolates from 8 different herds), serovar 9 (11 isolates from 5 herds), and serovar 16 (seven isolates from six herds). Other serovars were sporadic or absent (Appendix A).

### 3.2. Macrorestriction Clusters

A total of 16 clusters could be differentiated among the examined *A. pleuropneumoniae* strains, and the serovars differed markedly in their macrorestriction diversity (Appendix A). The dominant serovar 2 was highly diverse, with eight distinguishable clusters (G1 to J), featuring a major important cluster I2 including 29 isolates (more than half of all serovar 2 isolates) from at least 14 different herds. Serovar 13 biotype 2 was markedly less diverse, forming two potentially related subclusters with 14 and 11 isolates each from four herds (A1 and A2, respectively). All other major serovars (serovars 9, 10, and 16) were represented by a single cluster (clusters C, D2, and D1, respectively) showing high genetic relatedness. All major genetic clusters, except cluster D1, had a relatively long history of presence in Hungary; the first isolates of clusters A1, A2, C, D2, and I3 dated back to 1995, 1998, 2002, 2002, and 1998, respectively. All these major clusters have been shown to be present in the same herd for years (Figure 1).

### 3.3. Resistance Genes

Resistance against beta-lactam agents was frequently found in cluster A2; 4/11 isolates representing two herds were resistant to penicillin G (MIC = 64 mg/L in all cases) and amoxicillin (MIC = 128 mg/L in all cases), while 2/11 were also resistant to cefoperazone (MIC = 16–32 mg/L). The four penicillin and amoxicillin-resistant isolates carried the gene *bla*_ROB-1_. Isolates of cluster A2 were also frequently resistant to oxytetracycline (6/11 representing two herds; MIC = 16–128 mg/L) and to doxycycline (5/11; MIC = 4–8 mg/L). All doxycycline-resistant isolates harbored the *tet*L gene. The single oxytetracycline-resistant but doxycycline-susceptible isolate exhibited a relatively low minimum inhibitory concentration (MIC) of 16 mg/L and harbored neither *tet*L nor any other tested tetracycline resistance gene. Resistance against tetracyclines was also frequent in cluster A1; 6/14 isolates were resistant against oxytetracycline and doxycycline (MIC ranges as for cluster A2). Five of these isolates (from the same herd) carried a *tet*L gene and *tet*B was present in the remaining isolate. A single sporadic isolate harbored a *tet*B gene (Appendix A).

Regarding biotype 1 isolates, a single isolate in cluster C (serovar 9) carried a *tet*H gene (oxytetracycline MIC = 16 mg/L). In cluster D (serovar 16), one isolate harbored a *tet*B (oxytetracycline MIC = 16 mg/L) and another a *tet*L gene (oxytetracycline MIC = 32 mg/L). Isolates of cluster F carried a *tet*B gene (oxytetracycline MIC = 16 mg/L); two isolates from different herds in cluster I1 carried *tet*L genes (oxytetracycline MIC = 64 mg/L). Of the isolates of the large I2 cluster, one showed decreased susceptibility to penicillin G (MIC = 4 mg/L), one a *tet*B (oxytetracycline MIC = 8 mg/L), and another one a *tet*L gene (oxytetracycline MIC = 64 mg/L) (Appendix A). Other isolates were susceptible to the drugs tested; beta-lactamase genes *bla*_ROB-1_, *bla*_BRO-1_/*bla_BRO-2_* were not detected.

### 3.4. Toxin Profile

All isolates carried the species-specific *apxIV* gene, confirming species identification, but major differences were detected in toxin gene carriage patterns between biotypes, serovars, and clusters (Appendix A). All biovar 2 isolates (serovar 13 clusters A1, A2, and serovar 14 cluster B) lacked the genes *apxIII* and *apxIA*, carrying the genes *apxII* and *apxIB*, indicating the ability for the production of ApxII toxin only. Biovar 1 serovar 2 isolates generally carried *apxIB*, *apxII*, and *apxIII* (clusters G-I), indicating the ability for the production and secretion of ApxII and ApxIII, but not ApxI, associated with high virulence. Biovar 1 serovar 2 isolates of cluster J carried only the genes *apxII* and *apxIBD* and were capable of producing and secreting only ApxII, similarly to biotype 2 isolates. Sporadic biotype 1 isolates (serovar 12) behaved similarly, i.e., carried only *apxII* and *apxIBD*. In contrast, clusters C (serovar 9), D1 (serovar 16), D2 (serovar 16), and E (untypeable isolates) carried the genes *apxIA*, *apxIBD*, and *apxII*; thus, these isolates are likely to produce and secrete the toxin ApxI, associated with high virulence, together with ApxII.

### 3.5. Whole-Genome Analysis

The genome sizes were 2.26–2.99 Mbps and the GC contents were 41.03–41.26% (Appendix A). Genome completeness was >98.2%. The genomes contained 2055 core and 181 shell (i.e., present in >15% and <95% of isolates) genes; variability is limited in both the core genome sequence variability (proportion of variable sites 0.004) and presence/absence of accessory genes. Based on their core genome, the nine isolates sequenced were split into two distinct groups; however, short branch lengths indicate shallow genetic differentiation within clades (Figure 2). Raw reads are available at NCBI as Bioproject PRJNA874317. 

The group of isolates A149, A16, and A12 exhibits a large deletion in the region associated with type IV pilus assembly, lacking the genes *tadB*, *tadA*, *tadZ*, *rcpB*, *rcpA*, *rcpC*, *tadV*, *flp2*, *flp1*, and *rhlB*. These also harbor the broad-host range phage Salmon118970 Sal3. All nine genomes harbor the Mannheimia phage vB_MhS_587AP2. Resistance genes, apart from those already mentioned, were not found.

## 4. Discussion

The 16 PFGE clusters showed the high variability of *A. pleuropneumoniae* strains isolated in Hungary; however, whole-genome sequencing indicated well-conserved genomes that differ mainly in phage insertions, capsule polysaccharides, lipopolysaccharides, and RTX clusters that encode serotype-specific antigens [58,59].

The serovar distribution of *A. pleuropneumoniae* is variable in the different geographical regions. In Europe, generally, serovars 2, 4, 7, 8, and 9 are dominant [30,31,60,61,62]; however, serovar 8 was found to be the most frequent one in the UK [32]. Serovar distribution in this study of Hungarian isolates showed similarity to data from other European countries; however, serovar 1, formerly reported as frequent [63,64], disappeared, and the high proportion of serovar 13 (biotype 2), as well as serovar 16 (biotype 1), was also characteristic [29]. This seems to be a country-specific phenomenon, as in the Czech Republic, which is geographically close to Hungary, these serovars 13 and 16 were not reported [30,31]. In such cases, the differences cannot be explained convincingly by only the geographical distribution of strains. Nevertheless, the factors in the background of these differences are largely unknown. Potential explanations include (i) locally frequent serovars remain frequent because they have the highest chance to be introduced to a susceptible herd; (ii) virulence properties, i.e., highly virulent strains/serovars, are found more frequently; (iii) vaccination patterns, as bacterin vaccines offer protection against certain but never against all serovars; (iv) clonal advantages, i.e., differences in resistance, or in predisposition to the development of resistance, to antimicrobials used to treat porcine respiratory diseases or to disinfectants used in farms; (v) interaction with other infectious agent(s) that are involved in the etiology of the porcine respiratory disease complex; and (vi) technological differences, e.g., differential susceptibility of pig breeds farmed or differences in containment procedures.

Virulence properties probably contribute strongly to differences in serovar frequency. Those serovars producing the most potent ApxI together with ApxII (serovar 1, 5a, 5b, 9, and 11) are among the most frequently isolated serovars in many geographical regions. Similarly, serovars producing the cytotoxic but not hemolytic ApxIII together with ApxII (serovar 2, 4, 6, 8, and 15) are also frequently found. Thus, the relatively high frequency of serovar 9, which is a regionally important serovar [30,31], may be attributed to its high presumed virulence. It should be noted that all of the isolates tested in the present as well as in the cited Czech studies originated from pathological cases and none of them were isolated from asymptomatic carriers.

As serovar 16 has been shown to produce ApxI together with ApxII [24], its importance in Hungarian herds may be explained in the same manner; the high prevalence of serovar 16, at present, seems to be specific to the Hungarian epidemiological situation. Isolates of serovars 9 and 16 were highly clonal, the same clones were found in several geographically and technologically distinct farms, and persisted consistently over time in Hungary, lending further support to the above presumption.

Similarly, the higher frequency of serovar 2 in Hungary—as well as in other European countries [28,29,65]—than in North America [3] may also be caused by the production of one additional toxin (ApxIII) by the European strains. However, the virulence properties inferred from toxin production patterns do not fully explain the dominance of the probably less-virulent serovars 2 or 13 over serovars 9 and 16.

Serovar 2, in contrast to any other major serovar found, was highly diverse, characterized by four major clusters comprised of eight subclusters (G1-3, H, I1-4, and J). This is perfectly in line with previous findings [66], where ribotyping, as well as PFGE, showed a degree of diversity among *A. pleuropneumoniae* serovar 2 isolates, not only in Hungary but also in other European countries and in Canada. Similar results were obtained using amplified fragment length polymorphism in Danish isolates [67].

Cluster J was confined to a single farm and showed a toxin gene pattern similar to American serovar 2 isolates, i.e., *apxIBD*, *apxII* and a lack of *apxIII*, thus producing only ApxII besides the universal ApxIV. Consequently, these are probably less virulent than the other Hungarian serovar 2 isolates, which sufficiently explains their geographically and temporally limited presence and may represent an importation event without long-lasting persistence. In contrast, the three other major clusters were found in abundance throughout the country, with an overwhelming dominance of subcluster I2 present in 14 of the 44 geographically isolated farms sampled. This dominant cluster showed the same toxin gene profile as the other less-frequent subclusters within cluster I or within serovar 2. The dominance of serovar 2 in Hungary may be explained by a local/regional effect of integrated pig husbandry firms supplying pigs to multiple farms. However, this does not fully explain the diversity patterns of this serovar, especially the overwhelming dominance of subcluster I3 among serovar 2 isolates.

Serovar 13 was the second most frequent serovar in this study, which seems to be country-specific. Biotype 2 strains are more commonly isolated in Europe, e. g., in Hungary [29,35], Denmark [36], and Spain [37], than in the US [68].

These Hungarian serovar 13 isolates carried an additional *apxIII* toxin gene, unlike other characterized serovar 13 isolates [37,69]. Though the production of Apx*I* was also reported earlier in atypical serovar 13 isolates [69], it seems to show limited spread. In contrast, Hungarian serovar 13 isolates proved to be highly clonal: all isolates except one clustered together in two related subclusters. This conservation was confirmed by the whole-genome sequencing of representative isolates. Such high clonality is usually characteristic of highly virulent serovars such as serovars 9 and 16 in this study or serovars 1 and 5 in Canada, where the diversity of these serovars was found to be limited, especially considering isolates from animals with disease [70]. In the case of our serovar 13 isolates, two major differences were detected between the two groups of isolates. One was a carriage of a phage with a broad host range, in line with earlier reports of integrated phages as sources of variability [58]. The other difference involved the deletion of multiple genes of the type IV pilus assembly gene cluster, indicating the deficient expression of this important adhesin. Based on isolation dates, the long persistence of both groups is documented in Hungary.

Some serovar 13 isolates representing multiple farms showed resistance against tetracyclines as well as to certain beta-lactams. Resistance against these agents was markedly less frequent in other clusters. The high prevalence of resistance against these drug groups is documented in several different countries [71,72,73,74,75], which is not surprising, as these drug families are among the drugs used most frequently for prophylaxis and metaphylaxis in swine [76,77], as well as for growth promotion, where the use of antibiotics for such purposes is not banned. The tetracycline resistance genes detected were similar to those found by Blanco et al. [42], though *tet*O was not found.

Isolates from the same herd always showed similar resistance profiles and carried the same tetracycline resistance gene. However, isolates of the same (sub)cluster originating from different herds did not necessarily behave similarly and sometimes carried different tetracycline resistance genes, suggesting the acquisition of resistance multiple times and a lack of clones with stable drug resistance. Tetracyclines and beta-lactams are widely used for the treatment of pigs in Hungary, though herd-specific consumption data were not available. Antibiotic resistance, consequently, probably contributed minimally or not at all to the distribution patterns of most serovars, and the appearance of resistant strains is most probably driven by local antibiotic usage patterns. This is supported by the low level of resistance in the most frequent pulsotype I3.

The effect of vaccination on the present serovar distribution is probably limited, as vaccines were used in only a few farms before 2010 and the bacterin-type vaccines used offered protection only against serovars 1 and 2. The lack of serovar 1 isolates may be associated with bacterin vaccines, though this presumption is weakened by the disappearance of this serovar well before the widespread use of the vaccines. Serovar 2, in turn, remained the most frequent serovar in spite of the availability of the vaccine.

Though the increase in the frequency of serovar 13 coincided with the introduction of porcine circovirus (PCV) and porcine reproductive and respiratory virus (PRRSV) into the Hungarian pig industry (unpublished observations), systematic studies have not yet been performed to assess their predisposing role. Upcoming studies remain to be performed to assess whether there are differences between *A. pleuropneumoniae* serovars in their capability to cause disease synergistically with other pathogens of the porcine respiratory disease complex.

The contribution of wild boars to the spread is negligible, as all farms are closed farms, thus the risk of contact of pigs with wild boars is minimal [6].

As detailed data on the technological differences between herds shown to harbor *A. pleuropneumoniae* are not available, technological causes favorable for certain serovars cannot be unequivocally ruled out.

## 5. Conclusions

*Actinobacillus pleuropneumoniae* strains isolated from pigs in Hungary could be assigned into 16 clusters when examined with pulsed-field gel electrophoresis. The results confirm the high variability of the species; some serovars showed unusual toxin gene profiles and the frequent carriage of tetracycline resistance genes. The unique serovar distribution in Hungary is probably multifactorial. Virulence differences explain the high frequency of several but not all serovars. The occurrence of other infectious agents or management factors may also play a role in the spread of certain serovars or clones. Their impact must be analyzed in the future.

## Figures and Tables

**Figure 1 vetsci-09-00511-f001:**
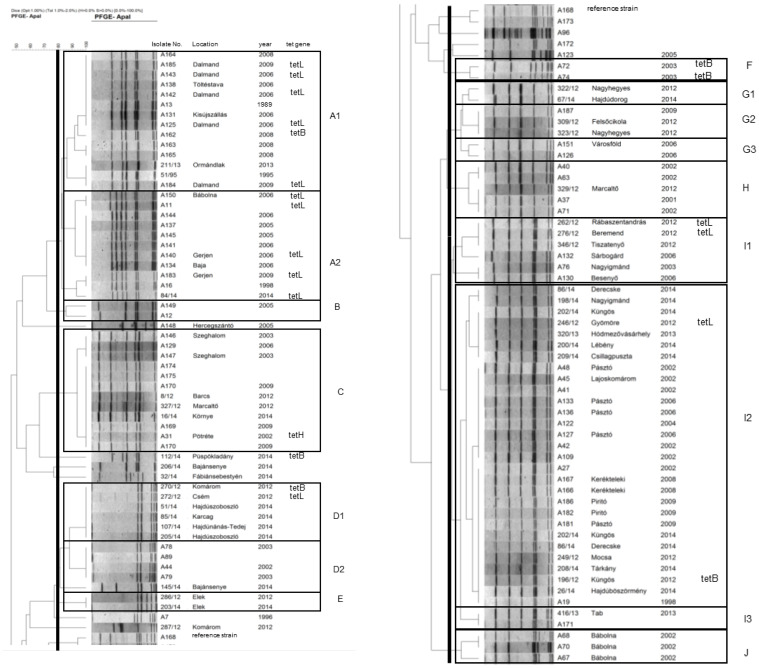
Dendrogram of *Actinobacillus pleuropneumoniae* isolates based macrorestriction by pulsed-field gel electrophoresis. The dendrogram was split and set side by side at the reference strain (A168) shown in both parts.

**Figure 2 vetsci-09-00511-f002:**
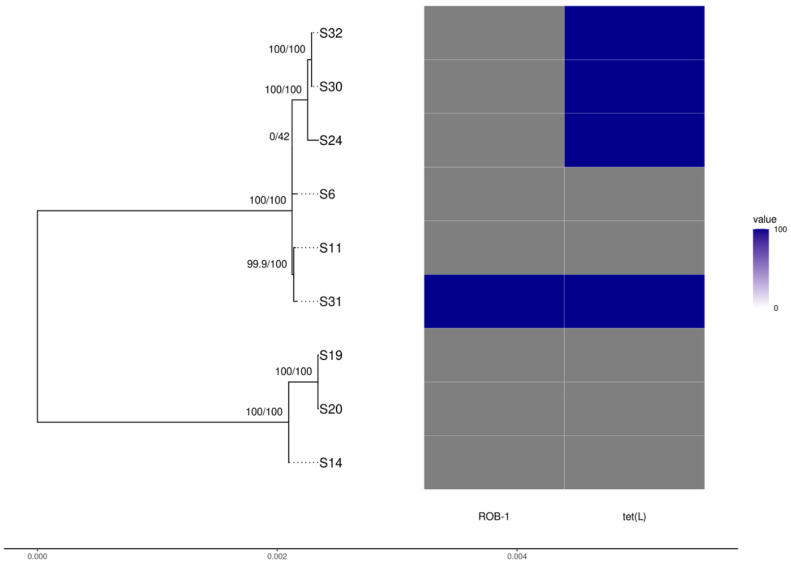
Phylogenetic tree of *Actinobacillus pleuropneumoniae* isolates reconstructed using the core genome alignment with the presence/absence of antibiotic resistance genes found.

## Data Availability

Raw reads are available at NCBI as Bioproject PRJNA874317. The majority of data generated during this study are included in this published article and its Appendix A.

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
