# Peer review of "Genetic Diversity of Actinobacillus pleuropneumoniae Serovars in Hungary"

_vetsci, 2022, doi:10.3390/vetsci9100511_

Round 1
Reviewer 1 Report
The manuscript by Kardos et al., reports the genetic characteristics of Actinobacillus pleuropneumoniae isolated in Hungary, classifying them into different serovars and also describing the antibiotic resistance pattern and toxin production genes.
The study is well designed and executed and the manuscript is well written.
Comments:
1) accession numbers for none of the samples from this study were found in the manuscript. The whole genomes need to be submitted to Genbank and accession numbers provided here.
2) conclusion should mention about the antibiotic resistance and toxin gene.
Author Response
The authors thank for the comments and suggestions of Reviewer 1. We fully agree with your comments, the text has been modified accordingly.
Specifically,
- Raw reads are available at NCBI as Bioproject PRJNA874317. Whole genomes have been uploaded and are under processing by the curators.
- A sentence on the correlation between antibiotic resistance and toxin genes has been added.
Reviewer 2 Report
This manuscript from Kardos et al reports the results of genetic sequencing done to detect different strains of A. pleuropnemoniae in necropsied pigs from 44 different swine herds in Hungary. This work is relevant considering the burden A. pleuropnemoniae infection puts on the swine industry worldwide and its integration of antibiotic resistance genes. Following bacterial culture, strains of A. pleuropnemoniae were identified by PCR and pulsed field gel electrophoresis, and genetic relatedness was determined using whole genome sequencing. 16 clusters were identified with these methods, with differences among them concerning genetic relatedness and diversity and their inclusion of antibiotic resistance and exotoxin genes. Notable findings include an apparent longstanding history of these bacterial strains in Hungary dating at least back to the 1990's and evidence of differentiation by geographic location, with the strongest genetic similarities being found among the same herds. As a whole, the paper is straightforward and well-written. I have two comments to address prior to acceptance for publication:
1) The authors should clarify that the electrophoresis gels shown in Figure 1 are showing the results of restriction digest and not PCR. It would also be useful for the authors to include PCR results in a separate figure, particularly because the antibiotic resistance genes can't be identified with the data shown in Figure 1 despite being listed.
2) Considering that the antibiotic resistance genes also showed some level of herd-specific differentiation, the authors should indicate whether these findings correlate with antibiotic usage in these various herds, if the information is available.
Author Response
The authors express their sincere thanks to Reviewer 2.
Regarding the suggestions
- We modified the legend to clearly state that the figure is based on macrorestriction. We would refrain from adding a figure showing PCR products on gel for the sake of conciseness. Four different resistance genes were found (tetB, tetH, tetL and blaROB), meaning that four gel photos would be needed.
- Yes, tetracyclines and beta lactams are widely used for treatment of pigs in Hungary, but no data are available for drug use in the specific farms involved. A sentence regarding this issue has been inserted into the text.
Reviewer 3 Report
A manuscript entitled “Genetic diversity of Actinobacillus pleuropneumoniae serovars in Hungary” by Kardos et al is describing about 116 Actinobacillus pleuropneumoniae strains isolated in Hungary from 1995 to 2014. Serovar 2 (50/116), serovar 13 (25/116), serovar 9 (11/116) and serovar 16 (7/116) were identified as the major prevalent serovars by the serological study. Those isolates were subdivided into 16 cluster by the pulsed-field gel electrophoresis. Among them 9 isolates were representatively subjected to the whole genome analysis. The results are almost properly analyzed. As the porcine pleuropneumonia caused by Actinobacillus pleuropneumoniae is one of the important diseases in the swine industry, the results presented in this manuscript can provide the useful information that may use to control this disease. Thus, this reviewer thinks this manuscript is worth to be published in Veterinary Sciences. However, before accepting the manuscript, a major and several minor points should be addressed by the authors.
Major point;
Character of 116 isolates, ie: biotype, serovar, cluster is not shown all together in any figures or tables, but a just a description. It is cruel for the reader to understand the features of 116 isolates obtained in Hungary. This reviewer suggests to add a new table or to redraw the figure 1 to show at least biotype, serovar, cluster so as to show the character of 116 isolates at once. Information regarding to the production of ApxI, II, III and IV, and the resistance to antibiotics such as tetracycline is also welcome.
Minor points;
1. Abstract, lines 22-24; The meaning of “in case of serovar 13” is not clear. Nine representative isolated were subjected to the whole genome analysis. According to Figure 1 and a sentence “serovar 13 clusters A1 and A2, serovar 14 cluster B” at lines 193-194, seven isolates (A16, A125, A140, A185, 82/14, 211/13, 51/95 ) belongs to serovar 13 and two isolates (A12, A149 ) belongs to serovar 14. Why the authors do not mention about two isolates in serovar 14.
2. Introduction, lines 77-78; The meaning of “a given region” is not clear. Make the meaning clearer.
3. Introduction, lines 82-83: “NAD-independent strains” should be “biotype 2 strains” from the definition of Actinobacillus pleuropneumoniae biotype.
4. Materials and Methods, subsection 2.1 Samples, lines 88-93; Terms that 119 strains were isolated should be shown as the information. Referring the figure 1, terms may be shown as 1995-2014.
5. Materials and Methods, lines 106; “Serotyping” had better be shown as “Serovar typing”.
6. Materials and Methods, lines 106; The meaning of “in a later phase” is not clear. Make the meaning clearer. Does it mean “using the liquid cultured bacterial cell in a late logarithmical phase”?
7. Materials and Methods, lines 112; Confirm two words, blaROB and blaBRO, if or not they are properly shown, because the papers the authors referred show them as blaROB-1 and blaBRO1/blaBRO2, respectively. This comment is also applied for Results section, line 188.
8. Materials and Methods, subsection 2.4 Whole genome sequencing, lines 123-141; Show the reason why nine isolates are selected here or the results section 3.5.
9. Results, lines 159 and 161; Cluster “D” is shown here, but “D” is shown as “D1” and “D2” in figure 1. According the figure 1, “cluster C and D” (line 159) and “clusters A1, A2, C, D and I3” (line 161) should be “cluster C, D1 and D2” (line 159) and “clusters A1, A2, C, D1, D2 and I3”, respectively.
10. Results, figure 1 legend, line 164. “Figure 1” is redundant. Delete one of two.
11. Results, figure 1 legend, lines 166-167. A sentence “The dendrogram was split at the reference strain (A168) shown in both part” had better be described as “The dendrogram was split and set side by side at the reference strain (A168) shown in both part.”
12. Results, figure 1; Vertical bold lines at the both sides of left picture may be an unnecessary noise. Delete both lines.
13. Results, figure 1; Items showing year, tetracycline gene and cluster are not aligned properly. Show and set each item correctly.
14. Results, figure 1; Location where A168 strains were isolated is not shown properly.
15. Results, figure 1; Cluster of A68, A70 and A67 were shown as “J” in figure 1, but is shown as “J1” in Results section line 154. Unify the description.
16. Results, subsection 3.4 Toxin profile, lines 190-203; Toxin profile is only shown by description in the main text. This reviewer suggests the authors to add a new table showing the toxin profile of 116 isolates including the reference strain.
17. Discussion, line 246-248; A sentence “v) interaction with other infectious agent(s), i.e. differences in the ability to cooperate with other pathogens causing porcine respiratory disease complex,” is shown as one of the potential explanations for surviving several serovar of Actinobacillus pleuropneumoniae strains in the fields, but “the ability to cooperate with other pathogens” is not clear. Refine the sentence.
18. Discussion, line 281-283; A sentence “In contrast, the three other major clusters were found in abundance throughout the country, with an overwhelming dominance of subcluster I3 present in 14 of the 44 geographically isolated farms sampled.” is unclear. Clarify the sentence, because dominance of subcluster I3 cannot be read from figure 1.
19. Data Availability Statement, line 359; Why “Not applicable”? In “Instructions for Authors”, MDPI shows that MDPI is committed to supporting open scientific exchange and enabling authors to achieve best practices in sharing and archiving research data. This reviewer recommends to discuss about this issue with the editorial office of Veterinary Science.
Author Response
The authors thank Reviewer 3 for his comments and suggestions
Regarding the Reviewer’s points
Major point
We prepared a Supplementary Table as requested. This comment also led to detection of an unfortunate error; 114 instead of 116 isolates were included. This was also corrected.
Minor points
- The abstract sentence was modified. Serovar 14 was considered sporadic and mentioned as such.
- The sentence has been modified. It is of interest that the frequency of the different serovars in the different countries and regions shows great variability.
- Accepted, the text has been modified accordingly.
- Corrected as suggested.
- Modified as suggested.
- Isolates were identified and stored at -80 ° Serovar typing was carried out when a larger number of isolates was collected. “Later phase” meant that a later period in the study. We removed the ambiguous part.
- We modified the text as requested. We also added a sentence clarifying that four isolates were blaROB-1 positive.
- We added a sentence to the Methods to explain the reason for choosing the isolates for sequencing
- Yes, we mistakenly considered clusters D1 and D2 together as cluster D. We corrected the mistake.
- Deleted as suggested.
- A168 is a reference strain isolated outside of Hungary.
- The Table was added as a Supplementary Table.
- Accepted, the sentence has been modified.
- This was a mistake, thank you for pointing it out. The dominant cluster is I2. The text was modified.
- We fully agree, now we have deposited the raw reads at NCBI under Bioproject PRJNA874317. This was properly cited in the new version.
Round 2
Reviewer 3 Report
The manuscript (vetsci-1868185) is now revised almost properly by the authors according to the suggestions and comments.
This reviewer thanks the authors for the efforts of revision, but would like to propose three minor points before accepting this manuscript.
(1) Newly added supplemental Table S2 in MS-Ward file; A width of column for "A149" is narrow to show the "Total length" in a line.
(2) Line 365, Data Availability Statement; Please consider to add following sentence. "Majority of data generated during this study is included in this published article and its supplementary tables."
(3) Make "Supplementary Materials" section above "Data Availability Statement" section and describe as follow; The following supporting information can be downloaded at: www.mdpi.com/xxx/s, Table S1: Characteristics of isolates., Table S2: Genome contiguity statistics as output by quast.".
Author Response
The authors appreciate the Reviewer's careful attention and comments to better the manuscript and accept all three remarks gratefully.
Supplementary Table 2 was amended.
The data availability has been updated as suggested.
The Supplementary Material section has been added.